# An Update on the Management of Acute High-Risk Pulmonary Embolism

**DOI:** 10.3390/jcm11164807

**Published:** 2022-08-17

**Authors:** Romain Chopard, Julien Behr, Charles Vidoni, Fiona Ecarnot, Nicolas Meneveau

**Affiliations:** 1Department of Cardiology, University Hospital Besançon, 25000 Besancon, France; 2EA3920, University of Burgundy Franche-Comté, 25000 Besancon, France; 3F-CRIN, INNOVTE Network, 42055 Saint-Etienne, France; 4Department of Radiology, University Hospital Besançon, 25000 Besancon, France

**Keywords:** high-risk pulmonary embolism, systemic thrombolysis, surgical embolectomy, catheter-based therapy, multidisciplinary care team

## Abstract

Hemodynamic instability and right ventricular (RV) dysfunction are the key determinants of short-term prognosis in patients with acute pulmonary embolism (PE). High-risk PE encompasses a wide spectrum of clinical situations from sustained hypotension to cardiac arrest. Early recognition and treatment tailored to each individual are crucial. Systemic fibrinolysis is the first-line pulmonary reperfusion therapy to rapidly reverse RV overload and hemodynamic collapse, at the cost of a significant rate of bleeding. Catheter-directed pharmacological and mechanical techniques ensure swift recovery of echocardiographic parameters and may possess a better safety profile than systemic thrombolysis. Further clinical studies are mandatory to clarify which pulmonary reperfusion strategy may improve early clinical outcomes and fill existing gaps in the evidence.

## 1. Introduction

Pulmonary embolism (PE) is a relatively common disease, with an incidence ranging from 60 to 112 per 100,000 inhabitants of the United States [1], and is the third most common cause of death among patients with cardiovascular diseases [2]. PE with associated hemodynamic instability or cardiac arrest carries a high risk of early mortality according to the risk stratification proposed in evidence-based practice guidelines [3,4]. High-risk PE accounts for about 5% of the total number of PE presentations but is associated with a heavy prognostic burden [5]. Overall, in-hospital mortality ranges between 22.0% and 31.8%, increasing to 65% if cardiac arrest occurs [5,6,7,8]. The rate of major bleeding at 30 days can reach 24.0% [9]. Abrupt increases in pulmonary vascular resistance and right ventricular (RV) afterload through direct pulmonary vascular obstruction, hypoxemic vasoconstriction, and release of pulmonary vasoconstriction factors are the main mechanisms leading to cardiogenic shock and death in this critical situation. Early recognition and treatments tailored to each individual are crucial to prevent fatal hemodynamic collapse [3,4]. The present paper provides an evidence-based critical review of the management of acute high-risk PE and highlights areas that remain to be clarified or resolved by ongoing and future research.

## 2. Methods

We searched PubMed and the Cochrane database for publications from January 2000 through May 2022 of randomized clinical trials, meta-analyses, systematic reviews, and observational studies published in English (see the Appendix A for the detailed search terms). The search yielded the following results: 1025 PubMed citations, 508 in OVID and 373 in the Cochrane database. After excluding non-eligible study types and duplicates, we manually searched the reference lists of selected articles, reviews, meta-analyses, and practice guidelines. Selected articles were mutually agreed on by the authors. A total of 102 articles (14 randomized clinical trials, 21 systematic reviews and meta-analyses, 57 observational cohort studies, 1 review, and 9 guideline documents) were included in the final review. Randomized clinical trials, meta-analyses, and information of interest to a general medical readership were prioritized.

## 3. Risk Stratification and Early Diagnosis of High-Risk Pulmonary Embolism

### 3.1. Risk Stratification

The European Society of Cardiology (ESC) guidelines categorize acute PE as being at low, intermediate, and high risk of 30 days mortality based on hemodynamic status, bedside clinical scoring systems such as the Pulmonary Embolism Severity Index (PESI) and simplified PESI (sPESI) [10,11], RV function on imaging [12], and cardiac biomarkers (i.e., troponin and brain-type natriuretic peptide) [13,14] with a view to therapeutic decision making [3]. High-risk PE refers to PE with hemodynamic instability, which encompasses a wide spectrum of clinical conditions from cardiac arrest to sustained hypotension through cardiogenic shock, all typically associated with RV dysfunction and elevation of cardiac biomarkers in relation to pathophysiological patterns of severe PE (Figure 1).

### 3.2. Diagnostic Strategy

Bedside trans-thoracic echocardiography (TTE) is the most useful initial test in unstable patients and will yield evidence of RV dysfunction if acute PE is the cause of hemodynamic decompensation. Figure 2 displays echocardiographic features in the assessment of RV strain in acute PE. Mobile right-heart thrombus (RHT) was found to be related to a significant increase in short-term mortality in a meta-analysis of six relevant studies totaling 15,220 patients (16.7% mortality in those with RHT vs. 4.4% in those without; odds ratio (OR), 3.0; 95% CI, 2.2–4.1) [14].

In highly unstable patients, TTE evidence of RV dysfunction suffices to prompt immediate pulmonary reperfusion without further testing. Computed tomography pulmonary angiography (CTPA) should be performed in stable or stabilized patients for confirmation of diagnosis [3]. TTE also makes it possible to explore the differential diagnosis of hemodynamic instability [15]. D-dimer testing is not required in this urgent situation where there is already a very high pre-test probability according to TTE [16].

## 4. Multidisciplinary Response Team

A dedicated multidisciplinary response team merges the expertise and unique perspectives of various disciplines who have an interest in PE, making it possible to personalize care based on each patient’s risk profile, comorbidities, and preferences [17]. Some studies have shown modifications in PE management [18,19] and potential improvement in the prognosis of PE with use of multidisciplinary response teams [18,20].

## 5. Emergency Management of Acute High-Risk Pulmonary Embolism

### 5.1. Anticoagulant Therapy

Anticoagulation should be initiated promptly when PE is diagnosed or when a high clinical suspicion exists on TTE while awaiting CTPA confirmation of diagnosis [3,4]. Anticoagulant therapy aims to reduce mortality from PE, morbidity of thrombus extension, and recurrence. Greater awareness regarding the thrombotic risk of subtherapeutic anticoagulation associated with unfractionated heparin (UFH) has driven a growing preference for the more consistent antithrombotic effect of low-molecular-weight heparin (LMWH) (Table 1) [3,4]. A meta-analysis of randomized studies reported that LMWH reduced the risk of recurrent VTE and major bleeding in the initial treatment period (OR, 0.69; 95% CI, 0.49–0.98, and OR, 0.44; 95% CI, 0.19–1.01) as compared to UFH [21]. Moreover, UFH is associated with a higher incidence of heparin-induced thrombocytopenia vs. LMWH (0.2% with LMWH vs. 2.6% with UFH; OR, 0.47; 95% CI, 0.22–1.02) [22]. The currently recommended thrombolytic regimens for the treatment of acute PE were evaluated in combination with UFH given as an initial bolus dose followed by a continuous infusion. A shift to LMWH or direct oral anticoagulant (DOAC) was allowed early after completion of thrombolysis in recent clinical trials of thrombolytic therapy for acute PE [23,24]. Only limited and uncontrolled data are currently available about the use of thrombolytic treatment in association with LMWH, fondaparinux, or DOACs in PE patients, but all of them showed the safety of such approaches with systemic thrombolysis [25,26,27].

### 5.2. Hemodynamic and Respiratory Support

#### 5.2.1. Oxygen Therapy

Administration of supplemental oxygen using O_2_-saturation-adapted nasal cannula, high-flow oxygen, or mechanical ventilation is indicated in patients with SaO_2_ < 90%. When mechanical ventilation is required, care should be taken to limit its adverse hemodynamic effects. The positive intrathoracic pressure induced by mechanical ventilation may reduce venous return and worsen RV failure in patients with shock; therefore, positive end-expiratory pressure should be applied with caution. Low tidal volumes (approximately 6 mL/kg lean body weight) should be used in an attempt to keep the end-expiratory plateau pressure below 30 cm H_2_O [31].

#### 5.2.2. Modest Fluid Loading

Experimental studies have shown that aggressive volume loading may worsen RV function by causing mechanical overstretch and/or inducing reflex mechanisms that depress contractility [32]. A clinical study including 13 PE patients with low cardiac output revealed an increase in the cardiac index from 1.6 ± 0.1 to 2.0 ± 0.1 L/min/m^2^ (*p* < 0.05) after infusion of 500 mL of dextran over 20 min [33].

#### 5.2.3. Vasopressors or Inotropes

Use of vasopressors (i.e., norepinephrine, 0.2–1.0 µg/kg/min) and/or inotropes (i.e., dobutamine, 2–20 µg/kg/min) is often necessary in parallel with (or while waiting for) reperfusion therapy to sustain cardiovascular function by keeping systemic arterial pressure greater than pulmonary function [31]. In a canine model of acute obstruction of the pulmonary circulation, norepinephrine infusion restored mean arterial pressure to baseline, decreased biventricular filling pressure, and increased cardiac index [34]. Dobutamine has been reported to increase both cardiac index and stroke index and to reduce pulmonary vascular resistance in acute PE patients [35].

#### 5.2.4. Cardio-Pulmonary Resuscitation and Extra-Corporeal Membrane Oxygenation

In cardiac arrest presumably caused by acute PE, current guidelines for advanced life support, including cardiopulmonary resuscitation (CPR), should be followed [36]. Extracorporeal membrane oxygenation (ECMO) is increasingly used in patients during cardiac arrest and cardiogenic shock to support both cardiac and pulmonary function. In a meta-analysis of 21 studies and 636 high-risk PE patients, ECMO was indicated due to cardiac arrest in 58.7% and obstructive shock in 41.3% of patients. The pooled estimate of early all-cause mortality was 41.1%. The most common in-hospital adverse event was major bleeding, with an estimated rate of 28.6% [37].

### 5.3. Reperfusion Therapies

Reperfusion strategies for the treatment of high-risk PE include systemic thrombolysis, surgical embolectomy, and catheter-based therapy (CDT).

#### 5.3.1. Full-Dose Systemic Thrombolysis

International guidelines unanimously recommend standard-dose systemic thrombolysis as the mainstay of reperfusion therapy for acute high-risk PE to rapidly reverse hemodynamic compromise, RV dysfunction, and gas exchange abnormalities [3,4]. Three thrombolytic drugs have been approved for high-risk acute PE, namely streptokinase, urokinase, and recombinant tissue plasminogen activator (rtPA). Treatment regimens include a loading dose followed by continuous infusion, with infusion times ranging from 2 h (rtPA, streptokinase, and urokinase) to 12 to 24 h (streptokinase, and urokinase). rtPA may be used with an accelerated regimen (i.e., 15 min), which has not been officially approved but is sometimes used in extreme hemodynamic instability, such as cardiac arrest [3]. Absolute contraindications to thrombolytic therapy include history of stroke in the preceding 6 months; central nervous system neoplasm; major neurologic, ophthalmologic, abdominal, cardiac, thoracic, vascular, or orthopedic surgery or trauma (including syncope associated with head strike or skeletal fracture) within the previous 3 weeks; and active bleeding [3].

The indication for systemic thrombolysis in high-risk PE is based on a number of small-sized trials that first demonstrated hemodynamic improvement within minutes or hours of treatment using clinical outcomes and surrogate parameters, including total pulmonary resistance, the degree of angiographic resolution, and mean pulmonary artery pressure (Table 2) [38,39,40,41]. Overall, full-dose systemic thrombolysis significantly decreased the rate of PE-related mortality (OR, 0.15; 95%, 0.03–0.78) and the rate of death or treatment escalation (OR, 0.18; 95% CI, 0.04–0.79) compared to anticoagulation alone [42]. Recent data from a nationwide cohort in Germany showed that thrombolysis was associated with lower in-hospital mortality rates in patients with hemodynamic instability, both in those with shock not necessitating CPR or mechanical ventilation (OR, 0.42; 0.37–0.48) and in those who underwent CPR (OR, 0.92; 95% CI, 0.87–0.97) [8]. This was achieved at the cost of a 9.5% rate of major bleeding (vs. 3.5% with heparin) and a 2.0% rate of intracranial hemorrhage (ICH) (vs. 0.19% with heparin) in populations not limited to patients with high-risk PE [43]. Analyses of unselected patient data report a rate of thrombolysis-associated ICH ranging from 3% to 5% [44,45]. The high rate of major bleeding observed after the administration of systemic thrombolysis is currently considered acceptable given the particularly poor prognosis of high-risk PE if left untreated [3,4]. Moreover, nationwide data showed that, among unstable patients with acute PE, only 30% actually received recommended thrombolytic therapy [8,46].

#### 5.3.2. Half-Dose Systemic Thrombolysis

A recent propensity score–matched study comparing outcomes in 3768 patients receiving 50 mg versus full-dose 100 mg of alteplase for PE demonstrated that half-dose systemic thrombolysis was ineffective, with an increased frequency of treatment escalation (53.8% vs. 41.4%; *p* < 0.01), driven largely by secondary thrombolysis (25.9% vs. 7.3%; *p* < 0.01) and CDT (14.2% vs. 3.8%; *p* < 0.01), with similar rates of in-hospital mortality and ICH (13% vs. 15%, and 0.5% vs. 0.4%, respectively) [47].

#### 5.3.3. Catheter-Directed Therapy

The desire to minimize the risk of adverse events, especially ICH, or to offer alternatives to systemic full-dose systemic thrombolysis in patients with a high bleeding risk has driven the development of alternative strategies for pulmonary reperfusion with CDT, which includes ultrasound-facilitated catheter-directed thrombolysis and thromboaspiration.

The most extensively studied CDT technique is ultrasound-facilitated, catheter-directed thrombolysis with the EKOSonic system (Boston Scientific Corporation, Marlborough, MA, USA). The EKOSonic device makes it possible to infuse low-dose rtPA directly through the pulmonary clot, with concomitant ultrasound impulsions. The potential advantage of local delivery is the lower dose of thrombolytic agent required, with the goal of reducing bleeding events. In the ULTIMA (Ultrasound Accelerated Thrombolysis of Pulmonary Embolism) trial, 59 patients with PE and RV dysfunction were randomized to receive either heparin alone (n = 29) or heparin plus catheter-directed thrombolysis (10–20 mg rtPA) facilitated by ultrasound (n = 30) [48]. The primary endpoint, namely the difference in RV/LV ratio from baseline to 24 h, was significantly improved in the CDT group compared to heparin alone (0.30 ± 0.20 vs. 0.03 ± 0.16; *p* < 0.001). Several other non-randomized, non-comparative studies including hemodynamically unstable PE patients or hemodynamically stable PE with RV dysfunction found comparable results in terms of the improvement in RV function. Major bleeding rates ranged between 0% to 16.6% and ICH rates between 0% to 2.4% in these studies [49,50,51,52,53,54,55,56,57,58,59,60,61]. In the subsequent dose-ranging OPTALYSE PE trial, four accelerated-dosing regimens (8 mg/2 h, 8 mg/4 h, 12 mg/6 h, and 24 mg/6 h) for ultrasound-facilitated, catheter-directed thrombolysis were evaluated in 101 patients with intermediate-risk PE. Across all four arms, which used a shorter delivery duration and lower-dose rtPA, there was improved RV function and reduced clot burden compared with baseline. Major bleeding was observed in 4% of patients, with two ICH [62].

Two other CDT devices using a purely thromboaspiration principle have been evaluated in single-arm trials in terms of efficacy and safety for the endovascular management of PE and have also been authorized by the FDA. Thromboaspiration catheters function by exerting negative pressure without the concomitant use of thrombolysis. In the multicenter EXTRACT-PE study, the 8 French Penumbra Indigo^®^ Aspiration Catheter (Penumbra, Alameda, CA, USA) was tested in 119 patients with hemodynamically stable PE and associated RV dysfunction. The results showed a significant improvement in RV function, as evaluated by the RV/LV ratio, with a mean reduction of 27.3%. Two patients experienced major bleeding (1.7%), of whom one died [63]. The prospective, multicenter FLARE trial tested the FlowTriever^®^ device (Inari Medical, Irvine, CA, USA) for the management of intermediate-risk PE in 106 patients. The primary endpoint of the FLARE study was the change in RV/LV ratio from baseline to 48 ± 8 h or discharge, whichever occurred first. The results showed a significant mean reduction of 0.38 (25.1%, *p* < 0.001) in the primary efficacy endpoint. Mean pulmonary artery pressure declined on average from 29.8 mmHg pre-procedure to 27.8 mmHg post-procedure (*p* = 0.001). Four patients experienced major adverse events within 48 h of the procedure, including one major bleeding event and one pulmonary vascular injury requiring lower lobectomy [64]. However, no data in high-risk patients are currently available with these thromboaspiration catheters. Evidence-based clinical guidelines recommend that CDT should be considered for patients with high-risk PE, in whom thrombolysis is contraindicated or has failed (level of recommendation, IIa) [3].

#### 5.3.4. Surgical Embolectomy

Surgical pulmonary embolectomy is considered in patients with intermediate-high-risk or high-risk PE in whom fibrinolysis has failed or is contraindicated in patients with large, centrally located PE [3,4]. Optimal results are achieved when the patient is referred before the development of pressor-dependent hypotension or cardiogenic shock [65]. A 2017 meta-analysis of 56 studies found a post-operative in-hospital mortality rate of 26.3% (95% CI, 22.5–30.5%) with a significant improvement over time (37.2% vs. 19.0%) [66]. Interestingly, recent studies reported that in-hospital mortality among patients with PE undergoing surgical thrombectomy in dedicated PE centers was 3–6%, and the 1-year mortality rate was 15–23% [67,68].

## 6. Special Situations

### 6.1. High-Risk PE Associated with Pregnancy

PE is the sixth leading cause of maternal death in the United States [69]. High-risk PE during pregnancy requires careful assessment and management with a multidisciplinary team and the obstetric providers in charge. Systemic thrombolysis and surgical embolectomy are both valuable pulmonary reperfusion options in this setting [3,4]. A systematic review of studies including 127 antepartum and postpartum women treated with systemic thrombolysis reported maternal survival to be up to 94%, with an associated 28.4% risk of major maternal bleeding, mostly occurring after delivery due to vaginal hemorrhage or post-cesarean-section abdominal bleeding. Fetal or neonatal death is reported to be 12%, whereas 35.1% of pregnant women receiving thrombolytic therapy had a pre-term delivery [70]. Among 36 women managed with surgical thrombectomy, maternal survival and risk of major bleeding were 86.1% and 20.0%, respectively, with fetal deaths possibly related to surgery in 20.0% [70]. Finally, ECMO could be considered as a viable life-support modality for pregnant and postpartum patients with critical cardiac or pulmonary illness [71].

### 6.2. Acute PE with Underlying Chronic Thrombo-Embolic Pulmonary Hypertension

Chronic thromboembolic pulmonary hypertension (CTEPH) is characterized by persistent pulmonary arterial obstruction, pulmonary vasoconstriction, and a secondary small vessel arteriopathy, resulting in chronic dyspnea and functional limitation with potential poor prognosis if untreated. CTEPH occurs in 2% to 4% of patients in the first 2 years after a symptomatic PE event [72]. Early recognition of underlying CTEPH in acute PE is of importance for individual management, whereas the diagnosis of unknown, underlying CTEPH is often missed. Indeed, the median time between CTEPH symptom onset and diagnosis in expert centers was reported to be 14 months [73]. TTE features that should prompt suspicion of underlying CTEPH in acute management include peak tricuspid regurgitation velocity > 2.9 m/s, RV/LV basal ratio > 1.0, flattening of the interventricular septum, right outflow doppler acceleration time < 105 ms with mid-systolic notching, early diastolic pulmonary regurgitation velocity > 25 mm, pulmonary artery diameter > 25 mm, IVC diameter > 21 mm with decreased respiratory collapse, and right atrial area (end-systole) > 18 cm^2^. Figure 3 illustrates the CT-scan features of acute PE with and without underlying CTEPH.

The management of acute PE in such patients is based on systemic thrombolysis. A small decrease in pulmonary vascular obstruction may improve RV failure since beyond 80% of pulmonary obstruction, there is an exponential relationship between pulmonary vascular obstruction and pulmonary vascular resistance [74]. Nevertheless, patients with catastrophic clinical presentation may be referred to ECMO for hemodynamic support [75], followed by pulmonary thrombo-endarterectomy or balloon pulmonary angioplasty for the dedicated treatment of CTEPH [76,77]. In-hospital mortality in CTEPH patients with RV failure dropped from 31% in 2005–2013 to 4% in 2014–2019 (*p* = 0.03) since the implementation of ECMO in CTEPH management [75].

## 7. Current Controversies Warranting Further Research

### 7.1. Risk Stratification

Recent data suggest that early risk stratification of acute PE could be improved by taking account of additional covariates, especially in high-risk PE. Data from 784 consecutive PE patients prospectively enrolled in a single-center registry suggested that the use of an optimized venous lactate cut-off value (i.e., 3.8 mmol/L vs. 2.3 mmol/L with the current definition) to diagnose obstructive shock made it possible to distinguish risk of in-hospital adverse outcomes between patients with shock and persistent hypotension (21.4% vs. 9.5%, respectively), resulting in a net reclassification improvement (0.24 ± 0.08; *p* = 0.002) [78]. Moreover, we previously demonstrated that the addition of renal dysfunction on top of the current ESC risk stratification improved overall model performance, yielding 18% reclassification of overall predicted mortality, including reclassification across intermediate–high-risk to high-risk PE in 15.8% of eligible patients [79].

### 7.2. Pulmonary Reperfusion Strategy

The use of systemic thrombolysis is based on historical studies that included very few patients (50 patients in total across all studies) [24]. The rationale for the development of catheter-directed pulmonary revascularization techniques is based on the reduction in the risk of major bleeding, notably ICH. CDT techniques have all shown promising results [49,50,51,52,53,54,55,56,57,58,59,60,61]. However, the comparison of these techniques to a control group with a parenteral anticoagulant or to another pulmonary reperfusion strategy is lacking and remains needed. Currently, only 59 patients were included in the sole randomized study to test the EkoSonicTM system [48] compared to over 1700 patients randomized in studies testing systemic thrombolysis in all PE presentations [80]. Furthermore, the value of using ultrasound to facilitate in situ thrombolysis remains debated. The SUNSET (Standard vs. Ultrasound-Assisted Catheter Thrombolysis for Submassive Pulmonary Embolism) trial did not show any difference in pulmonary arterial thrombus reduction on CT scan (*p* = 0.76), among 81 patients randomized to receive in situ thrombolysis either with or without facilitated ultrasound impulsions [81]. The outcomes used to date to evaluate the different endovascular reperfusion devices in PE have been surrogate endpoints, whereas data with hard, clinically relevant endpoints, such as mortality, are lacking. In a meta-analysis of 16 studies totaling 860 patients treated with CDT, the mortality rate was 12.9% for high-risk PE and 0.74% in normotensive patients with PE and RV dysfunction. The rates of major bleeding and ICH were 4.65% and 0.35%, respectively [82]. In a German nationwide inpatient cohort, based on administrative data without pre-specified analysis, CDT was associated with lower in-hospital mortality rates compared to systemic thrombolysis (OR, 0.30; 95% CI, 0.14–0.67), with an associated 1.2% rate of ICH in the CDT group [83]. The ongoing HI-PEITHO trial will assess whether ultrasound-facilitated, catheter-directed thrombolysis is associated with a reduction in the composite outcome of PE-related mortality, cardiorespiratory decompensation, or nonfatal PE recurrence compared to anticoagulation alone in intermediate-high risk PE [84]. No such RCT in high-risk PE vs. systemic thrombolysis, for instance, is currently registered.

The administration of systemic thrombolysis in refractory cardiac arrest potentially related to PE still remains debated [85,86]. A meta-analysis including 4384 all-comers with cardiac arrest did not show an improvement in hospital discharge rate with systemic thrombolysis during CPR compared to conventional management (3.5% vs. 10.8%; risk ratio, 1.13; 95% CI, 0.92–1.39; *p* = 0.24, I^2^ = 35%) [85]. The pulmonary reperfusion approach between surgical embolectomy, thrombolysis, and anticoagulation alone in high-risk PE requiring an ECMO life support also remains unclear [3,87,88]. We reported in a meta-analysis of 17 studies and 327 patients that mechanical reperfusion, notably by surgical embolectomy, yields favorable results as compared to other reperfusion strategies (OR, 0.43; 95%CI, 0.23–0.997; *p* = 0.009; I^2^ = 35.2%) regardless of the timing of ECMO implantation in the reperfusion timeline and independently of thrombolysis administration or cardiac arrest presentation, with similar rates of bleeding events [89].

#### 7.2.1. Failure of Systemic Thrombolysis

Thrombolysis is unsuccessful in up to 8% to 12% of high-risk PE patients [90,91]. Guidelines recommend performing surgical embolectomy or CDT if thrombolysis has failed, based on a low level of evidence [3,4]. We previously reported an unadjusted rate of in-hospital death of 7% in patients treated with rescue embolectomy versus 38% in those receiving a repeat thrombolysis among 40 acute PE patients [90].

#### 7.2.2. Right-Heart Thrombus

Right-heart thrombus (RHT) is detected in <4% of patients with PE [92] and is associated with worse outcomes [93]. Given the limited existing evidence, guidelines or consensus statements have not addressed the management of RHT [3,4,94]. An observational pooled analysis of 328 cases of RHT in transit suggested that thrombolysis (OR, 4.8; 95% CI, 1.5–15.4) and surgical embolectomy (OR, 2.6; 95% CI, 0.9–7.6) were associated with a more favorable outcome than anticoagulation alone [95]. Among 18,803 patients from the RIETE registry, including 443 with RHT, a propensity-matched analysis (82 pairs) did not find any reduction of all-cause death (OR, 0.86; 95% CI 0.30–2.43) or PE-related death (OR, 0.65; 95% CI 0.20–2.16) in patients with RHT treated mainly with thrombolysis as compared to those not receiving reperfusion therapy [96].

## 8. Conclusions

High-risk PE still remains associated with a poor prognosis despite advances in our understanding of the pathophysiological processes and prognostic factors. Urgent multidisciplinary care is crucial to manage this life-threatening situation. Systemic thrombolysis is the cornerstone of pulmonary reperfusion, but CDT has shown promising results. Nevertheless, the overall treatment of acute high-risk PE is based on a low level of evidence, and numerous clinical situations remain unresolved. Additional data from large registries or ideally RCTs are mandatory to better define the management of high-risk PE and improve patient prognosis.

## Figures and Tables

**Figure 1 jcm-11-04807-f001:**
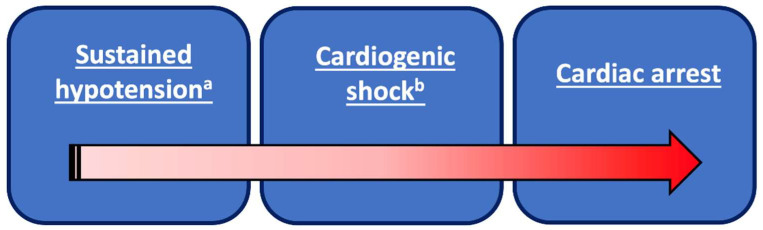
Clinical spectrum of high-risk pulmonary embolism. (a) Systolic blood pressure (BP) < 90 mmHg or systolic BP drop ≥ 40 mmHg, lasting longer than 15 min, and not caused by new-onset arrhythmia, hypovolemia, or sepsis; (b) systolic BP < 90 mmHg or vasopressors required to achieve a BP ≥ 90 mmHg despite adequate filling status and end-organ hypoperfusion (altered mental status; cold, clammy skin; oliguria/anuria; increased serum lactate > 2.4 mmol/L).

**Figure 2 jcm-11-04807-f002:**
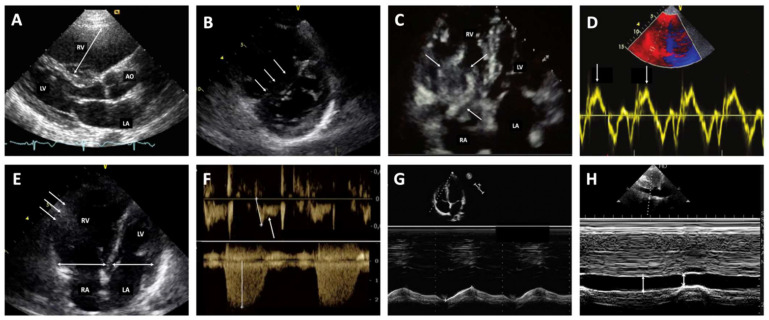
Trans-thoracic parameters for the assessment of right ventricular dysfunction in the acute phase of pulmonary embolism. (**A**). Enlarged right ventricle (parasternal long axis view). (**B**). Flattened intraventricular septum (arrows) (parasternal view). (**C**). Mobile thrombus (arrows) in the right heart cavities. (**D**). Decreased peak systolic (S’) velocity of tricuspid annulus < 9.5 cm/s (Tissue Doppler imaging). (**E**). Dilated right ventricle with basal RV/LV ratio > 1.0 (double-ended arrows), and McConnell sign (i.e., akinesia of the mid free wall (arrows) with normal motion at the apex hypokinesia of the RV) (four chamber view). (**F**). 60/60 sign: Association of acceleration time of pulmonary ejection < 60 ms and midsystolic “notch” with mildly elevated (<60 mmHg) peak systolic gradient at the tricuspid valve. (**G**). Decreased tricuspid annular plane systolic excursion (TAPSE) < 16 mm (M-Mode). (**H**). Distended inferior vena cava with diminished inspiratory collapsibility (subcostal view). RV, right ventricle; LV, left ventricle; Ao, aorta; RA, right atrium; LA, left atrium.

**Figure 3 jcm-11-04807-f003:**
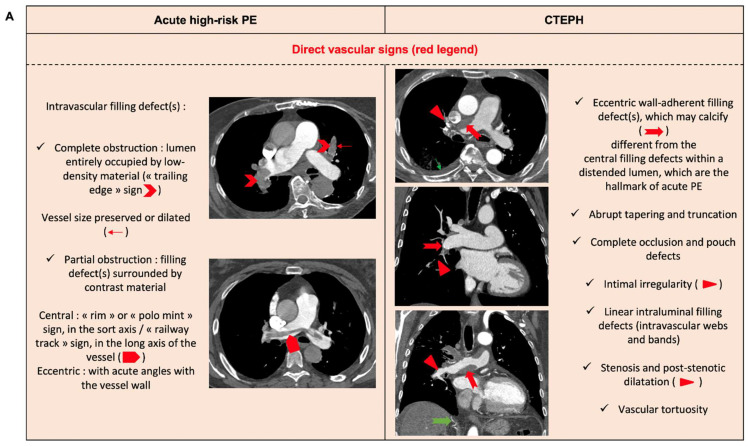
Direct vascular signs (**A**), indirect vascular signs (**B**), and parenchymal changes (**C**) on computed tomography scan between acute pulmonary embolism and chronic thrombo-embolism pulmonary hypertension (CTEPH).

**Table 1 jcm-11-04807-t001:** Anticoagulant options for the management of acute high-risk pulmonary embolism.

Drug	Dosage	Mechanism of Action	Efficacy	Adverse Effects	Practical Issues
Unfractionated heparin	80 IU/kg bolus followed by 18 IU/kg per hour by continuous infusion	Inhibitor of thrombin and factor Xa through an antithrombin-dependent mechanism	All-cause death within the first 10 days of unstratified VTE (19.0% PE): 0.0% (vs. 0.6% with LMWH) (n = 708) [28] Recurrent VTE within the first 10 days of unstratified VTE (19.0% PE): 0.3% (vs. 0.6% with LMWH) (n = 708) [28]	Major bleeding within the first 10 days of unstratified VTE (19.0% PE): 1.1% (vs. 1.4% with LMWH) (n = 708) [28]	aPTT ratio maintained between 1.5 to 2.0 per normal value. No issue with renal failure.
Low-molecular-weight heparin			All-cause death within the first 10 days of unstratified VTE (19.0% PE): 0.6% (vs. 0.0% with UFH) (n = 708) [28] Recurrent VTE during the initial within the first 10 days of unstratified VTE (19.0% PE): 0.6% (vs. 0.3% with UFH) (n = 708) [28]	Major bleeding within the first 10 days of unstratified VTE (19.0% PE): 1.4% (vs. 1.1% with UFH) (n = 708) [28]	To be reduced in case of renal failure. No evidence for dose adjustment based on coagulation tests.
-Enoxaparin SC	1.0 mg/kg every 12 h or A.5 mg/kg once per day				
-Tinzaparin SC	175 IU/kg once per day				
-Dalteparin SC	100 IU/kg every 12 h or 200 IU/kg once per day				
-Nadroparin SC	86 IU/kg every 12 h or 171 IU/kg once per day				
Fondaparinux	Once per day: 5 mg (body weight < 50 kg); 7.5 mg (body weight 50–100 kg); 10 mg (body weight > 100 kg)	Synthetic pentasaccharide that inhibits factor Xa	All-cause death at 3-month FU in unstratified VTE (19.0% of with PE): 0.8% (vs. 1.1% with UFH) (n = 2213) [29] Recurrent VTE within the first 7 days (19.0% of with PE): 1.3% (vs. 1.7% with UFH) (n = 2213) [29]	Major bleeding within the first 7 days in unstratified VTE (19.0% of with PE): 1.3% (vs. 1.1% with UFH) (n = 2213) [29]	Avoid in case of renal failure. No evidence for dose adjustment based on coagulation tests.
Argatroban (primarily in patients with suspected or confirmed HIT)	Initial: 2 mcg/kg/min IV continuous infusion over 1–3 h until steady state. Not to exceed infusion rate of 10 mcg/kg/min	Specific and reversible direct thrombin inhibitor	Thrombosis: 5.8% and 6.9% any new thrombosis at 30 days in HIT patients (vs. 15.0% and 23.0% in historical control groups) (n = 177 and n = 328) [30]	Major bleeding: between 3.1% and 5.3% at 30 days in patient with HIT (compared with between 8.2% and 8.6% in historical control groups) (n = 177 and n = 328) [30]	aPTT ratio maintained between 1.5–3 times initial baseline value. Check aPTT and adjust dose until target aPTT is achieved. To be reduced in case of renal failure Use caution in hepatic impairment.
Bivalirudin (primarily in patients with suspected or confirmed HIT)	Initial: 0.15–0.2 mg/kg/h IV;	Specific and reversible direct thrombin inhibitor	Thrombosis: 4.6% any new thrombosis at 30 days in patients with HIT (no comparator) (n = 461) [30]	Major bleeding: 7.6% at 30 days in patient with HIT (no comparator) (n = 461) [30]	Adjust to aPTT 1.5–2.5 times baseline value. Use caution in renal impairment: IV infusion, -Moderate (CrCl 30–59 mL/min): 1.75 mg/kg/h -Severe (CrCl < 30 mL/min): 1 mg/kg/h Hemodialysis: 0.25 mg/kg/h

SC, subcutaneously; IU, international units; aPTT, activated thromboplastin time; FU, follow-up; VTE, venous thrombo-embolism; UFH, unfractionated heparin; LMWH, low-molecular-weight heparin; HIT, heparin-induced thrombocytopenia.

**Table 2 jcm-11-04807-t002:** Characteristics and results of randomized clinical trials that evaluated systemic thrombolysis in pulmonary embolism, including high-risk patients.

	No. of Patients	Eligibility	Severity Criteria	Thrombolysis	High-Risk PE (%)	Main Results
UPET (1970) [41]	160	Acute PE < 5 days	Yes	Urokinase 12 h	8.7%	Rapid improvement of RV function and pulmonary reperfusion. Death within 2 weeks: 7.3% in the urokinase group vs. 8.9% in the heparin group. Recurrent VTE within 2 weeks: 15.3% in the urokinase group vs. 18.3% in the heparin group. Severe bleeding within 2 weeks: 45.0% in the urokinase group vs. 27.0% in the heparin group [41].
Ly (1978) [38]	25	Acute PE < 5 days	>1 lobe	Streptokinase 72 h	100.0%	Significant improvement in the mean pulmonary angiographic score with streptokinase 10.3 ± 5.1 vs. 3.7 ± 7.2 with heparin. Clinical improvement: 80.0% in the streptokinase group vs. 33.3% in the heparin group. Death: 10.0% the streptokinase group vs. 33.3% in the heparin group Major bleeding: 40.0% in the streptokinase group vs. 33.3% in the heparin group [38].
Dotter (1979) [40]	31	Acute PE	>1 lobe	Streptokinase 2–11 MIU 18–72 h	6.5%	The mean angiographic score-improvement rating was 2.08 in streptokinase-treated patients and 0.86 in heparin-treated patients Mean PA pressure changes in both groups were similar (streptokinase −7.9 mmHg, heparin −6.2 mmHg) [40].
Jerjes-Sanchez (1995) [39]	8	Acute PE < 14 days	Massive	Streptokinase 1.5 MIU/2 h	100.0%	The mortality in the streptokinase group was 0% compared with 100% (*p* = 0.02) in the heparin group [39].

RA, right atrial; RV, right ventricle; PA, pulmonary arterial; VTE, venous thrombo-embolism; MIU million international units.

## Data Availability

Not applicable.

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
