# Peer review of "An Update on the Management of Acute High-Risk Pulmonary Embolism"

_jcm, 2022, doi:10.3390/jcm11164807_

Round 1
Reviewer 1 Report
This paper tried to review the published articles about risk stratification, early diagnosis, management, reperfusion therapies of high-risk pulmonary embolism. This paper has been well organized and might have some value.
1. In 5.1, LMWH and novel oral anticoagulants are safer and more effective than UFH. One of the reasons is that they have a low rate of induction of heparin-induced thrombocytopenia (HIT); One of other reasons is that LMWH and novel oral anticoagulants have a lower risk of bleeding. These conclusions should be described in the text.
2. Throughout the whole paper, many paragraphs are too long to read, for example, the paragraph 7.1. In addition, the central idea of these paragraphs is not very clear.
Author Response
This paper tried to review the published articles about risk stratification, early diagnosis, management, reperfusion therapies of high-risk pulmonary embolism. This paper has been well organized and might have some value.
Thank you for your positive appreciation and useful comments for improvement.
1.In 5.1, LMWH and novel oral anticoagulants are safer and more effective than UFH. One of the reasons is that they have a low rate of induction of heparin-induced thrombocytopenia (HIT); One of other reasons is that LMWH and novel oral anticoagulants have a lower risk of bleeding. These conclusions should be described in the text.
Thank you for these pertinent remarks. We have modified paragraph 5.1 accordingly, except that we did not mention novel oral anticoagulants, because they are not indicated in the acute management of high-risk PE.
2.Throughout the whole paper, many paragraphs are too long to read, for example, the paragraph 7.1. In addition, the central idea of these paragraphs is not very clear.
Thank you for your assessment. We have shortened the manuscript throughout to make the main points clearer.
Reviewer 2 Report
Thank you for the possibility to review the manuscript titled: “An update on the management of acute high-risk pulmonary embolism”. The manuscript is interesting and easy to read, provides several important points for diagnosis and management of patients with PE.
-The introduction is short and can benefit from supplemental data. Particularly, the data on worldwide burden of PE and epidemiological data.
-The methods section can be supplemented with a flow chart (optional)
-Please review the dosages of all drugs. My particular concern is that inotropes (epinephrine, norepinephrine, dobutamine are give in grams (g). I believe that at this point the dosage was misspelled, therefore, please change the g dose into mcg.
-Table 2 is large and has multiple mistakes. Please correct the mistakes and make it more readable (smaller).
-Figure 2 provides a forest plot from a meta-analysis. However, it is hard to understand where did this figure appear from. There is no citation of the original study. Moreover, if the original figure is used please provide indication that it is reprinted with permission.
The manuscript is interesting, well-illustrated and discusses a “hot” topic. Please take into account the recommendations in the spirit of improving the quality of the submission.
Author Response
Thank you for the possibility to review the manuscript titled: “An update on the management of acute high-risk pulmonary embolism”. The manuscript is interesting and easy to read, provides several important points for diagnosis and management of patients with PE.
-The introduction is short and can benefit from supplemental data. Particularly, the data on worldwide burden of PE and epidemiological data.
Thank you for this suggestion. We have included additional citations regarding the worldwide burden of PE and epidemiological data in the introduction.
-The methods section can be supplemented with a flow chart (optional)
The full details of the search strategy are given in the Supplementary Material. Since the numbers of each article type are given in the text, we do not feel that a flowchart is necessary.
-Please review the dosages of all drugs. My particular concern is that inotropes (epinephrine, norepinephrine, dobutamine are give in grams (g). I believe that at this point the dosage was misspelled, therefore, please change the g dose into mcg.
Thank you for bringing this to our notice, this has now been corrected.
-Table 2 is large and has multiple mistakes. Please correct the mistakes and make it more readable (smaller).
Thank you for this suggestion. The Table has now been reduced in size and corrected.
-Figure 2 provides a forest plot from a meta-analysis. However, it is hard to understand where did this figure appear from. There is no citation of the original study. Moreover, if the original figure is used please provide indication that it is reprinted with permission.
The Reviewer is correct. We have therefore chosen to delete this figure to avoid any ambiguity.
The manuscript is interesting, well-illustrated and discusses a “hot” topic. Please take into account the recommendations in the spirit of improving the quality of the submission.
Thank you for your positive appreciation and useful comments for improvement. We hope that the revised version will now meet your satisfaction.